# Strategies for Anticancer Treatment in p53-Mutated Head and Neck Squamous Cell Carcinoma

**DOI:** 10.3390/biomedicines13051165

**Published:** 2025-05-10

**Authors:** Bi-He Cai, Chia-Chi Chen, Yu-Te Sung, Yu-Chen Shih, Ching-Feng Lien

**Affiliations:** 1School of Medicine, I-Shou University, Kaohsiung City 82445, Taiwan; 2Department of Pathology, E-Da Hospital, Kaohsiung City 82445, Taiwan; sasabelievemydream@gmail.com; 3Department of Plastic Surgery, E-Da Hospital, Kaohsiung City 82445, Taiwan; coldfire1021@gmail.com; 4Department of Otolaryngology-Head and Neck Surgery, E-Da Hospital, Kaohsiung City 82445, Taiwan; intuition430@gmail.com

**Keywords:** p53, mutation, head and neck cancer, HNSCC

## Abstract

This Opinion summarizes the strategies for anticancer treatment in p53-mutated head and neck squamous cell carcinoma (HNSCC). It examines six strategies for anticancer treatment in p53-mutated HNSCC: 1. direct reactivation of mutated p53; 2. activation of p63; 3. activation of p73; 4. degradation of mutated p53; 5. blocking the p53-regulated oncogenic microRNA; and 6. blocking the p53-regulated oncogenic long non-coding RNA. Since HNSCC has a high p53 mutation rate compared to other types of cancers, these strategies for combating p53-mutated HNSCC may prove useful for generating new ideas or methods for developing treatments for other cancers with p53 mutations. This article also explores other factors that may impact the effectiveness of anticancer therapies in p53-mutated HNSCC.

Head and neck cancers (HNSCCs) have the fourth-highest p53 mutation rate among all cancer types (Figure 1).

Various strategies have been developed to target p53-mutated HNSCC cells (Figure 2):Direct reactivation of mutant p53: APR-246 restores the DNA-binding activity of p53 mutants, reestablishing its function [2]. This compound induces cell death in FaDu HNSCC cells (with a p53 heterozygous R248L mutation) [3].Activation of p63: p63, a member of the p53 family, is rarely mutated in cancers [4]. Lovastatin activates p63, inducing cell death in FaDu HNSCC cells (p53 heterozygous R248L mutation) [5]. Doxorubicin enhances the binding of the c-Jun transcription factor to the TAp63 promoter, increasing p63 expression [6,7]. This response occurs in both HN30 cells (wild-type p53) and UMSCC10B cells (p53 heterozygous G245C mutation) [8]. Knocking down p63 partially reverses doxorubicin-induced apoptosis in UM-SCC10B cells, demonstrating that p63 is crucial for this apoptotic response [8].Activation of p73: p73, another p53 family member, also has a low mutation frequency in cancers [9]. Chlorophyllides activate p73, leading to apoptosis in Detroit 562 cells (p53 homozygous R175H mutation), as well as in TW01 and HONE-1 cells (both p53 heterozygous R280T mutation) [10]. RETRA and NSC59984 also activate p73, inducing cell death in Detroit 562 cells [11]. Additionally, miRNA-193a-5p inhibition can trigger p73 activation in JHU-029 HNSCC cells (p53 heterozygous G108Vfs*15 mutation) [12].Degradation of mutated p53: The chaperone proteins HSP40 and HSP90 stabilize mutated p53 [13,14]. DNAJA1, a member of the HSP40 family [15], supports mutant p53 stability. Its knockdown in CAL33 (p53 homozygous R175H mutation) and HN31 cells (p53 heterozygous C176F mutation) reduces p53 levels and impairs oncogenic traits such as cell migration and colony formation [14]. The HSP90 inhibitor AUY922 sensitizes FaDu (p53 heterozygous R248L mutation) and CAL-27 (p53 homozygous H193L mutation) cells to cisplatin [16].Blocking mutant p53-regulated oncogenic microRNAs: Mutant p53 induces the overexpression of oncogenic miR-182-5p [17]. Inhibiting miR-182-5p reduces cell proliferation and migration in CAL-27 cells (p53 homozygous H193L mutation) [18].Blocking mutant p53-regulated oncogenic long non-coding RNAs: The long non-coding RNA MIR205HG is a downstream effector of mutant p53 [19]. Knocking down endogenous mutant p53 in CAL-27 cells reduces lncMIR205HG expression. Antisense targeting of MIR205HG impairs colony formation in CAL-27 and FaDu cells [19].

Other factors may influence the response to anticancer drugs in HNSCC cancer cells carrying p53 mutation.

Specific single-nucleotide polymorphism (SNP) in p53, p63, and p73 genes.

The *TP53* gene harbors a well-characterized SNP, rs1042522, within its coding region [20]. This SNP results in a substitution at codon 72, where the G allele encodes arginine (R72), and the C allele encodes proline (P72) [21]. Several studies have reported that p53-mutated cells carrying the R72 variant exhibit relatively higher resistance to chemotherapeutic agents such as cisplatin and doxorubicin, compared to those with the P72 variant [22,23]. The variants of SNP sites rs17506395 and rs35592567 on the p63 gene could influence p63 expression amounts [24,25,26], so drug responses of p63 activators may differ in p53-mutated cells containing different alleles on rs17506395 or rs17506395. p73 twin polymorphism at bases 4 and 14 of p73 exon 2 (G4C14-A4T14) polymorphism could influence p73 expression amounts [27]. G4C14-A4T14 locates upstream of the translation start site to affect the translation of p73 to cause A4T14 to contain a much higher expression of p73 than G4C14 [28]. In addition, in one study, HNSCC patients carrying the p73 variant AT allele at the G4C14-A4T14 site were less likely to develop second primary malignancies compared to those with the p73 variant GC allele [29]. Therefore, drug responses of p73 activators may differ in p53-mutated cells containing different alleles on G4C14-A4T14 of p73.

2.
**Aggregation type of p53 mutations**


Two specific p53 mutations, R175H and R280T, have been shown to form prion-like aggregates in HNSCC cells [10,11,30]. These aggregated forms of mutant p53 can sequester p63 and p73, thereby inhibiting their tumor-suppressive functions [10,11,30].

3.
**Homologous or heterologous p53 mutation**


p53 and RB are both crucial tumor suppressor proteins that play key roles in regulating the cell cycle and preventing uncontrolled cell growth [31]. Tumor formation requires the loss of function in both RB alleles, while a single defective p53 allele can promote cancer by reducing p53 gene dosage, creating a cellular environment more favorable for additional oncogenic changes [32]. The p63 activator lovastatin can induce cell death in FaDu HNSCC cells [5], which carry a heterozygous p53 R248L mutation. Further evaluation is needed to assess the anticancer effects of lovastatin in other HNSCC cell lines with homologous p53 mutations.

4.
**Two mutations in p53.**


Cells with two p53 mutations have been reported in colon and breast cancers, with a frequency of 7% (3 out of 43 cases) in colon cancer and 13% (31 out of 234 cases) in breast cancer [33,34]. Among the 149 HNSCC cell lines listed in the Handbook of p53 Mutations in Cancer Cell Lines (http://p53.free.fr/Database/Cancer_cell_lines/HB_cell_lines.html, accessed on 14 April 2025), 7% (10 out of 149) were found to harbor two p53 mutations. These cell lines include MDA-1386, MDA-1686, OSC-3, OSC-5, OSC-6, OSC-8, OSC-9, UM-SCC-14, UT-SCC-10, and UT-SCC-16.HPV+ infection in p53 mutation cells. The response of HNSCC cancer cells with two p53 mutations to anticancer drugs warrants further investigation.

5.
**HPV-positive HNSCC**


Human papillomavirus (HPV) and particularly its high-risk subtypes play a significant role in the development of both cervical and HNSCC cancers [35,36,37]. While mutations in the TP53 gene are commonly associated with tumorigenesis, HPV-positive cancers often follow a distinct oncogenic pathway [38]. In these cases, the viral E6 oncoprotein targets the p53 tumor suppressor protein for degradation, effectively inactivating its function without the need for genetic mutation [39]. In cervical cancers, TP53 mutations are significantly less prevalent in HPV-positive tumors compared to HPV-negative ones, highlighting the sufficiency of E6-mediated p53 degradation in promoting malignancy [40]. A similar pattern is observed in HNSCC, where HPV-positive tumors exhibit reduced p53 expression due to E6 activity [41]. However, TP53 mutations are still observed in a subset of HPV-positive HNSCC cases [42]. These mutations can compromise the tumor-suppressive activity of p53 or, in some instances, endow the protein with gain-of-function properties that actively promote tumor progression [43]. HNSCC cells harboring both HPV infection and TP53 mutations may exhibit increased resistance to anticancer therapies compared to HPV-negative cells with TP53 mutations alone.

This Opinion article provides an overview of various strategies for targeting mutant p53 in HNSCC, offering potential approaches that could also be relevant to other cancers with high p53 mutation rates. We have also discussed additional factors that may influence the response of p53-mutated HNSCC to anticancer therapies.

## Figures and Tables

**Figure 1 biomedicines-13-01165-f001:**
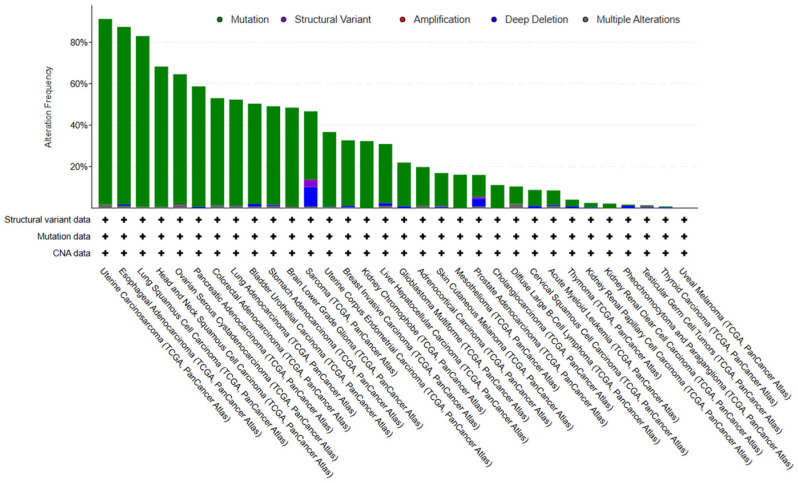
p53 mutation rates in different cancers. According to the cBioPortal database (cbioportal.org, accessed on 1 February 2025) [1], head and neck squamous cell carcinoma (HNSCC) has a p53 mutation rate of 68.26%, ranking fourth among all cancer types.

**Figure 2 biomedicines-13-01165-f002:**
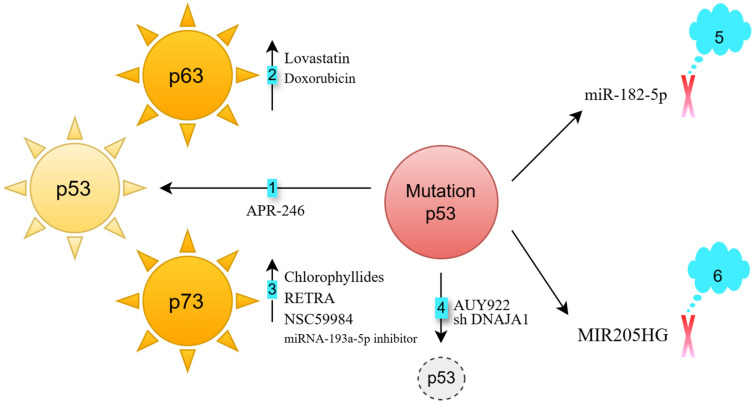
Strategies for targeting p53-mutated HNSCC. Six key strategies have been explored to target p53-mutated HNSCC. (1) Direct reactivation of mutant p53: APR-246 binds to and reactivates mutant p53, restoring its tumor-suppressor function. (2) Activation of p63: lovastatin and doxorubicin can activate p63 in p53-mutated HNSCC cells, leading to cancer cell death. (3) Activation of p73: chlorophyllides, RETRA, NSC59984, and the miRNA-193a-5p inhibitor activate p73 in p53-mutated HNSCC cells, inducing apoptosis. (4) Degradation of mutant p53: AUY922 and shRNA targeting DNAJA1 (shDNAJA1) promote the degradation of mutant p53, reducing oncogenic properties such as cell migration and colony formation in HNSCC cells. (5) Blocking mutant p53-regulated oncogenic microRNA: mutant p53 upregulates oncogenic miR-182-5p, inhibiting miR-182-5p reduces oncogenic properties, including cell proliferation and migration, in p53-mutated HNSCC cells. (6) Blocking mutant p53-regulated oncogenic long non-coding RNA: downregulation of the long non-coding RNA lncMIR205HG impairs multiple oncogenic properties, including cell proliferation, migration, and colony formation, in p53-mutated HNSCC cells.

## Data Availability

No new data were created or analyzed in this study. Data sharing is not applicable to this article.

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
