# Peer review of "Strategies for Anticancer Treatment in p53-Mutated Head and Neck Squamous Cell Carcinoma"

_biomedicines, 2025, doi:10.3390/biomedicines13051165_

Round 1
Reviewer 1 Report
Comments and Suggestions for Authors
In the article entitled “Strategies for anticancer in p53 mutated Head and neck squa-2 mous cell carcinoma”, the authors summarizes the strategies for anticancer treatment in p53-mutated Head and Neck Squamous Cell Carcinoma (HNSCC). In their opinion, they reported six strategies for combating p53-mutated HNSCC, suggesting them as useful for generating new ideas or methods for developing treatments for other cancers with p53 mutations.
The informations reported in this report could be interesting, and the manuscript is simple to read, but it seems to lack many informations about the current state-of-the-art about p53 research.
I have a concern about the informations reported:
For each strategy, the authors reported the presence of a p53 mutant. Due to the high frequency of p53 mutations in this tumour (Head and neck cancers (HNSCC) is reported as the fourth highest p53 mutation rate among all cancer types), are these strategies designed for more than the single p53 mutation reported?
In conclusion, data may be interesting but they have to be broader reported. I ask MAJOR REVISIONS for this manuscript, since the authors have to address my concerns as well as those from the other reviewers before to see their manuscript accepted for publication.
Author Response
In the article entitled “Strategies for anticancer in p53 mutated Head and neck squa-2 mous cell carcinoma”, the authors summarizes the strategies for anticancer treatment in p53-mutated Head and Neck Squamous Cell Carcinoma (HNSCC). In their opinion, they reported six strategies for combating p53-mutated HNSCC, suggesting them as useful for generating new ideas or methods for developing treatments for other cancers with p53 mutations.
The informations reported in this report could be interesting, and the manuscript is simple to read, but it seems to lack many informations about the current state-of-the-art about p53 research.
I have a concern about the informations reported:
For each strategy, the authors reported the presence of a p53 mutant. Due to the high frequency of p53 mutations in this tumour (Head and neck cancers (HNSCC) is reported as the fourth highest p53 mutation rate among all cancer types), are these strategies designed for more than the single p53 mutation reported?
A: We have added new sections discussing factors that may influence the response to anticancer drugs in HNSCC cells with p53 mutations. The presence of dual p53 mutations in HNSCC is also highlighted in the updated content.
In conclusion, data may be interesting but they have to be broader reported. I ask MAJOR REVISIONS for this manuscript, since the authors have to address my concerns as well as those from the other reviewers before to see their manuscript accepted for publication.
A: Thanks for suggestion.
Reviewer 2 Report
Comments and Suggestions for Authors
This short manuscript summerises possible approaches to reactivation of mutant p53 in cancer cells. It succeeds in this and is worthy of publication.
Two minor points should be addressed:
The authors should mention in their references to the effects of various agents on p53 mutated cells that different mutations could behave differently.
The word 'comprehensive' should be omitted from line 79-it is only a brief summary of the subject.
Author Response
This short manuscript summerises possible approaches to reactivation of mutant p53 in cancer cells. It succeeds in this and is worthy of publication.
Two minor points should be addressed:
The authors should mention in their references to the effects of various agents on p53 mutated cells that different mutations could behave differently.
A: We have added new sections addressing factors that may influence the response to anticancer drugs in HNSCC cells harboring p53 mutations. The updated content also covers the aggregation types of p53 mutations, whether the mutations are homologous or heterologous, and the presence of dual mutations in p53.
The word 'comprehensive' should be omitted from line 79-it is only a brief summary of the subject.
A: We deleted this word. Thanks for suggestion.
Round 2
Reviewer 1 Report
Comments and Suggestions for Authors
In the 2nd submission of the article entitled “Strategies for anticancer in p53 mutated Head and neck squamous cell carcinoma”, the authors responded to my comments.
With respect to my comments, they responded to the arised questions.
I think the manuscript may be considered for publication in its present form.